# Retinal Nerve Fiber Layer Thickness and Higher Relapse Frequency May Predict Poor Recovery after Optic Neuritis in MS Patients

**DOI:** 10.3390/jcm8112022

**Published:** 2019-11-19

**Authors:** Clara Grazia Chisari, Mario Damiano Toro, Vincenzo Cimino, Robert Rejdak, Maria Luca, Laura Rapisarda, Teresio Avitabile, Chiara Posarelli, Konrad Rejdak, Michele Reibaldi, Mario Zappia, Francesco Patti

**Affiliations:** 1Department “GF. Ingrassia”; section of Neurosciences. University of Catania, 95123 Catania, Italy; clarachisari@yahoo.it (C.G.C.); lucmaria@tiscali.it (M.L.); laura_rapisarda90@yahoo.it (L.R.); mario_zappia@yahoo.com (M.Z.); fpatti@outlook.com (F.P.); 2Department of General Ophthalmology, Medical University of Lublin, 20079 Lublin, Poland; robertrejdak@yahoo.com; 3Eye Clinic, University of Catania, 95123 Catania, Italy; t.avitabile@unict.it (T.A.); mreibaldi@libero.it (M.R.); 4IRCCS Centro Neurolesi “Bonino Pulejo”, 98124 Messina, Italy; vincenzo.cimino@irccsme.it; 5Department of Surgical, Medical, Molecular Pathology and of Critical Area, University of Pisa, 56126 Pisa, Italy; chiaraposarelli@gmail.com; 6Department of Neurology, Medical University of Lublin, 20079 Lublin, Poland; krejdak@yahoo.com

**Keywords:** acute optic neuritis, ocular coherence tomography, multiple sclerosis, retinal nerve fiber layer

## Abstract

Optic neuritis (ON) is a common manifestation of multiple sclerosis (MS). Aiming to evaluate the retinal nerve fiber layer (RNFL) with optical coherence tomography (OCT), patients with relapsing-remitting (RR) MS experiencing ON were consecutively enrolled. RNFL, ganglion cell layer (GCL), foveal thickness, and macular volume were evaluated in both the ON and unaffected (nON) eye within six days from the relapse onset (T0) and after six months (T1). Ninety patients were enrolled. At T0, ON eyes showed a significantly increased RNFL when compared to the nON eyes (129.1 ± 19.5 vs. 100.5 ± 10.1, *p* < 0.001). At T1 versus T0, the ON eyes showed a thinner RNFL (129.1 ± 19.5 vs. 91.6 ± 20.2, *p* < 0.001) and a significantly decreased GCL (80.4 ± 8.8 vs. 73.8 ± 11.6; *p* < 0.005). No differences were found in the nON group in retinal parameters between T0 and T1. A multivariate logistic regression analysis showed that a higher number of relapses (not ON) and a greater swelling of RNFL at T0 were associated with poor recovery. The assessment of RNFL through OCT during and after ON could be used to predict persistent visual disability.

## 1. Introduction

Axonal damage represents a major contributor to permanent neurological disability in several neurological diseases, such as multiple sclerosis (MS) [1,2]. Hence, understanding the dynamics and processes involved in axonal injury is fundamental for revealing the pathogenesis of brain disorders and developing biomarkers and therapies. Optic neuritis (ON) is one of the most common acute manifestation of MS, related to the inflammatory and neurodegenerative damage that could potentially affect the whole visual system, from the retina to the cortex [3,4,5]. During ON, acute inflammation damages axons at the retrobulbar level and this injury may extend to the body of the retinal ganglion cell layer (GCL) [6]. Retrograde axonal degeneration of GCL in MS patients with ON determines residual optic nerve head pallor and thinning of the peripapillary retinal nerve fiber layer (RNFL), as demonstrated in pathological [7] and imaging [8] studies of the retina. Lesions in the posterior part of the optic nerve, namely, retrobulbar neuritis [9], can occur without any detectable abnormality of the optic nerve head. On the other hand, the inflammation of the optic nerve close to the lamina cribrosa (the anterior limit of myelination) may lead to papillitis, characterized by oedema of the optic disc, blurring of the disc margin, and hyperemia [10]. 

Optical coherence tomography (OCT), a noninvasive and reproducible technique, investigates the retinal structures and the first unmyelinated part of the optic nerve through high-resolution tomographic sections [11,12]. This technique contributes to a better understanding of the neuroaxonal injury in ON while allowing the development of imaging markers that could be also used for clinical decision making [13]. Analysing morphology, reflectivity, thickness, and volume, OCT provides measurements of the retinal damage linked to the inflammatory demyelinating process during the acute phase and informs the clinical evaluation at follow-up [14,15]. Since the axonal degeneration linked to ON represents a suitable model to study the inflammatory processes typical of MS [11,14], an OCT analysis could provide insights about the events occurring during the acute phase of central nervous system (CNS) inflammatory diseases. 

This study aimed to analyze the retinal layers during the acute phase of ON MS relapses, and to evaluate axonal damage after six months. Moreover, it also attempted to identify predictors of permanent visual loss based on the OCT analysis.

## 2. Materials and Methods

### 2.1. Study Population

This observational, prospective study screened 121 patients newly diagnosed with RR-MS referring to the Neurologic Clinic of the University Hospital “Policlinico-Vittorio Emanuele” of Catania, in the period between 1st May 2013 and 31st March 2017. This study had been previously approved by the local ethics committee (Catania 1). All patients gave written informed consent. Patients with a diagnosis of “clinically definite MS” (McDonald’s diagnostic criteria 2010) [16] experiencing ON symptoms (neurologic visual symptoms lasting at least 24 h, without fever or other manifestations related to other pathologies [17]) were enrolled. Patients with myelitis suggestive of neuromyelitis optica, history of previous ON relapse, or previous ophthalmological issues such as optic disc pathology, glaucoma, lens cataract, corneal opacity, or with a grade of myopia greater than 5.5 diopters, or patients previously subjected to intraocular surgery or ocular laser treatment, were excluded from the study. Transient fever-related worsening of visual symptoms were not considered as ON relapses.

### 2.2. Clinical Assessment

After a detailed anamnesis, all the enrolled patients underwent a clinical, radiologic, and ophthalmological evaluation, as described below. In accordance with the criteria for ON retrieved from the Optic Neuritis Treatment Trial, patients with an acute progressive vision loss, possibly associated with pain on eye movement, visual field defects, and relative afferent papillary defect, were included in the study [18]. All patients were evaluated with visual evoked potentials. 

The clinical evaluation was performed by a highly experienced neurologist (F.P.). The assessment of clinical disability expressed by Expanded Disability Status Scale (EDSS) [19] was performed during relapse (T0) and after six months (180 ± 12 days) (T1). Data about disease duration, disease-modifying drug use, acute treatment with corticosteroids, and number of previous relapses (not ON) were also collected.

The radiologic evaluation included the brain MRI (magnetic resonance imaging) scan as standard clinical practice, in order to exclude other causes of visual acuity loss. 

Patients were examined during the acute phase (first symptoms occurring between 24–48 h before the clinical evaluation), before being treated with a high dose of corticosteroids (T0), and at follow-up after 6 months (180 ± 12 days from the relapse onset) (T1). The clinical improvement was evaluated through the *Delta* EDSS (ΔEDSS), the difference between EDSS at T1 and at T0. 

### 2.3. Ophthalmological Evaluation

An ophthalmological evaluation was performed by an highly experienced ophthalmologist (M.T.): the best corrected visual acuity (VA) for each eye was investigated with the Snellen eye test charts and the low-contrast Sloan letter acuity charts (LCSLA 100%, 2.5%, and 1.25%) at T0 and at T1. The decimal VA was converted to the logarithm of the minimal angle of resolution (logMAR) for statistical analysis, whereby logMAR = −logVA [20]. The difference between logMAR VA at T0 and logMAR VA at T1, expressed as a delta logMAR VA (ΔlogMAR VA), was used to assess visual recovery, assigning a poor recovery if the VA at T1 was +0.2 logMAR or greater (equivalent to <Snellen 6/9) [21]. Partial recovery was equaled to nonrecovery. 

OCT was performed with Stratus OCT (model Cyrrus 5000, Carl Zeiss Meditec, Dublin, CA). Peripapillary RNFL was acquired with the Optic Disc Cube 200 × 200 protocol that images the optic disc in a 6 mm × 6 mm region. The mean RNFL and that referred to individual quadrants were obtained. Macular GCL was obtained using the Macular Cube 512 × 128 protocol that images a 6 mm × 6 mm area centered at the fovea. The GCL was calculated automatically over an elliptical annulus (2 mm × 2.4 mm radius), excluding the central foveal region (0.5 mm × 0.6 mm radius). Macular volume (MV) and foveal thickness (FT) were also measured. Average measurements of three sequential circular scans of diameter 3.46 mm centered on the optic disc were recorded. The intereye differences in OCT measures (nonaffected eyes minus affected eyes) were also evaluated. A single operator blinded to the patients’ diagnosis collected all the measurements. Only well-focused and centered scans with a signal strength of >7 were included. Quality control and APOSTEL recommendations according to published criteria were followed [22,23]. No mydriatic drugs were used when performing the measurements. 

### 2.4. Statistical Analysis

Data were collected in a database created ad hoc and analyzed with the software STATA 11.0 [24]. The numerical datasets were tested for normal distribution with the Shapiro–Wilk test. Quantitative variables were expressed as mean and standard deviation. Mean and proportion differences between ON and non-ON (nON) eyes were evaluated with t-test and Chi square test, respectively. Nonparametric tests were used when appropriate. In order to stratify the RNFL swelling severity in ON eyes, we used 25% percentiles and median values as follows: grade 0 or no swelling (range 80.0–117.5 µm); grade 1 (117.6–127.0 µm); grade 2 (127.1–135.5 µm); grade 3 (>135.6 µm). To quantify the relationship between RNFL values and clinical parameters (VA, number of previous relapses), a linear regression analysis adjusted for the confounding factors (sex and age) was performed. A multivariate logistic analysis was conducted to evaluate the influence of clinical and retinal variables on the outcome (recovery yes/no). All the clinical and ophthalmological variables discussed above were included in the multivariate regression analysis as continuous covariates. Clinical outcome was the only categorized variable.

Independent clinical factors with a conservative significance level of *p* < 0.2 obtained in the univariate analysis were entered simultaneously into a multivariate model. The statistical significance was set at *p* ≤ 0.05.

## 3. Results

### 3.1. Clinical and Demographic Characteristics 

Demographic and clinical features are summarized in Table 1. Out of the 121 patients screened, 90 patients—with a mean age of 35.6 ± 10.5 years and a disease duration of 91.9 ± 85.2 months—were enrolled. Thirty-one patients did not meet the inclusion criteria. The mean EDSS was 3.11 ± 1.6 at T0 and 2.5 ± 1.8 at T1. 

Forty-seven patients (52.2%) had a complete postrelapse recovery (Table 1). The mean duration of visual symptoms was 35.9 ± 4.8 h (range 26.4–46.6). All patients were treated with steroids (methylprednisolone 1000 mg/die) intravenously administered for 5 days, as for protocol in our clinic. None of the patients was treated with immunosuppressive and/or immunomodulatory drugs during the observation period. 

### 3.2. Ophthalmological Evaluation and OCT Parameters

At T0, the ophthalmological evaluation showed that the ON eyes had a worse VA (calculated by LogMAR) and lower scores at LCSLA charts compared to the nonaffected ones. At T0, the OCT measurements showed that the ON eyes had thicker RNFL, with significantly higher value of retinal thickness for each quadrant, when compared to nON eyes (Table 2). 

At baseline, 7 (25.9%) patients showed no swelling (grade 0), 6 (22.2%) showed mild swelling (grade 1), 8 (29.6%) showed moderate swelling (grade 2), and 6 (22.2%) showed severe swelling (grade 3) in the ON eyes.

At T1, the ON eyes showed a significant improvement of VA and of all LCSLA charts’ scores (Table 3). The OCT analysis showed a significant reduction of the mean RNFL thickness and that of each quadrant when compared to T0. Moreover, GCL and MV were significantly decreased from T0 to T1 (Table 3; Figure 1A). 

Clinical and retinal parameters in the nON eyes are summarized in Table 3. Particularly, a slight but significant reduction of the mean RNFL and that regarding the nasal and temporal quadrants were observed between T0 and T1 (Table 3 and Figure 1B). 

Moreover, the mean intereye differences in RNFL, GCL, VA, and LCSLA were statistically significant between the two time points (Table 4). 

The correlation analysis showed that, in ON eyes, RNFL and VA were positively correlated at T0 (*r* = 0.89; *p* < 0.001) (Figure 2A). Moreover, RNFL at T0 and VA at T1 showed a positive correlation (*r* = 0.83; *p* < 0.005) (Figure 2B).

The multivariate regression analysis showed that the number of previous relapses (OR 3.54, 95% CI 1.82–3.84, *p* = 0.01), as well as a higher grade of RNFL swelling (OR 1.35, 95% CI 1.03–3.42, *p* = 0.05) were both independent predictors of a worse clinical outcome in patients with a visual relapse. In patients with both clinical factors (previous relapses and high grade of RNFL swelling), the OR raised to 5.12 (95% CI 3.56–8.62, *p* = 0.002).

## 4. Discussion

Our study demonstrated that retinal alterations in the ON patients followed a typical trend, characterized firstly by an increase of the RNFL thickness that decreased 6 months after the relapse. Several studies confirmed that retinal thickness swelling might be linked to the presence of oedema, due to the acute inflammatory process in ON [25,26,27,28], although two-thirds of these patients have showed a normal funduscopic examination [29,30,31]. A swelling of the inner nuclear layer and outer retinal layers in eyes with a history of acute ON has also been reported [32,33,34]. In addition, retinal oedema due to the involvement of the anterior visual pathway has been demonstrated in cases of retrobulbar optic neuritis [35] and could represent the consequence of the spreading of the interstitial oedema along the optic nerve [28]. Alternatively, the optic disc swelling could be exacerbated by the compression of the central retinal vein, especially in its retrolaminar portion, which consequentially reduces the drain of the oedema. Another explanation of the presence of retinal oedema is that a dysfunction of Mϋller cells may lead to an uncontrolled water clearance and, thus, to an increase of intercellular water in outer retinal layers [36]. Indeed, Mϋller cells, the most abundant glial cells in the retina, play a major role in maintaining the structural and functional stability of retinal layers [37]. Consequently, the noncystoid fluid accumulation may spread to the deeper retinal layers, thus, increasing their thickness. 

In particular, our findings may be explained by different pathological mechanisms combining inflammation-related dynamic fluid shifts and Mϋller cell dysfunction [38]. Accordingly, it has been reported that increased thickness of the inner nuclear layer (INL) and the outer plexiform layer was associated with disease activity in MS [27,39]. In a very recent longitudinal study, the increase in INL volume was found to be strongly associated with inflammation of the optic nerve [39]. 

In our study, the oedema may mask the axonal damage, that may become more evident after six months (Table 2; Figure 1A). A recent study demonstrated that, at the level of the optic nerve head, RNFL atrophy had occurred in the first 2 months and, although oedema was present in the optic disk at clinical onset, it disappeared after one month [31]. 

Moreover, it has been demonstrated that MS patients not experiencing ON show a significant retinal atrophy, confirmed by lower retinal thickness at OCT after six months; the same pattern is shared by subclinical ON or primary retinal degeneration and neurodegeneration processes [27,34,40,41,42]. In our study, retinal measurements of the unaffected eyes showed a significant reduction of RNFL thickness with particular involvement of the nasal and temporal quadrants (Table 2). In addition, in our study, OCT revealed an early reduction of the GCL, which was detectable at T0 and became more evident at T1 in both eyes, confirming that the retinal atrophic process, as previously demonstrated, can be immediately detected studying the GCL in both the ON and nON eyes [8,31,33,43,44].

To the best of our knowledge, this is the first study demonstrating that number of relapses (not ON) and RNFL swelling are predictors of poor recovery in MS patients with ON relapse. These results are consistent with other observations confirming the occurrence of a better postrelapse outcome in patients affected by idiopathic acute ON with a low grade of oedema and low visual impairment [28]. Marked swelling might be the expression of a more severe inflammatory process, which is responsible for a deeper RNFL alteration leading, in turn, to worse clinical manifestations, as supported by the statistical correlation between VA and RNFL at T0. Thus, it could be hypothesized that the autoimmune reaction and the release of cytokines and other molecules due to the inflammatory process might lead to a more severe damage of the axonal fibers, which could result in atrophy and in partial or no recovery in the long term [45]. Similarly, a clinical history of a high number of relapses could be interpreted as evidence of a more aggressive inflammatory disease. Accordingly, patients presenting both risk factors (swelling and elevated number of relapses) in our cohort had a 5-times greater OR of poor recovery. 

Compared to previous research, our study did not confirm the atrophy of GCL as a predictor of vision loss [31,33,46,47,48], probably due to the follow-up at 6 months. In fact, it has been demonstrated that OCT-measured structural changes could evolve for up to 12 months [49]. Moreover, we cannot rule out the possibility of missing the early GCL thickening, since the timing of the first OCT in our series varied from 2 to 6 days. Thus, a larger sample size and weekly OCT assessments in the first month from the ON onset may enable the clinician to better detect the early changes in the GCL and its influence on visual recovery. More recently, a multicenter international study identified the intereye difference of 5–6 μm in RNFL thickness as a robust structural threshold for identifying the presence of a unilateral optic nerve lesion in MS [50,51].

Compared to the available literature, our study did not report microcystic macular changes on OCT of the macular region in our cohort. However, the cause of these abnormalities remains unknown and the microcystic macular oedema was also detected in patients with non-MS ON and in several ophthalmological degenerative, with a large variation of prevalence, ranging from 0.8% to 25% [27,52,53]. 

This study has some limitations. Firstly, the short follow-up and the heterogeneity of patients in terms of disease duration, disability, and previous relapses; however, in order to limit heterogeneity, patients with previous ON were excluded from our study. Previous studies focused on acute ON; however, while the majority of them included patients with MS diagnosis and different phenotypes (RR-MS, progressive, and clinically isolated syndromes) [31,48,54,55], our study enrolled only newly diagnosed RR-MS patients with “pure” ON (with no previous ON relapse), in order to avoid a misinterpretation of the results. In fact, the retinal thickness was frequently demonstrated to be thinner in primary progressive MS, showing a correlation with disability, regardless to the presence of ON [56,57]. On the other hand, similarly to us, an already published research investigated a cohort of ON patients; however, swelling was not detected [54]. Secondly, the Frisen grade was not used to determine papilloedema severity in order to objectivize the optic disk evaluation and reduce the intrarater variability. Finally, brain and optic nerve MRI data were not collected; hence, future studies may focus on the possible correlation between brain and visual pathways imaging and retinal parameters.

## 5. Conclusions

In conclusion, our results demonstrated that RNFL swelling and the number of previous MS relapses may be able to predict clinical recovery in MS patients with ON in an early phase. The use of OCT in the routine evaluation of MS patients may allow to distinguish between patients having a greater chance to recover and those risking a worse outcome, who may benefit from a more aggressive therapeutic approach. In conclusion, the identification of patients at higher risk of poor recovery may guide the decision-making phase.

## Figures and Tables

**Figure 1 jcm-08-02022-f001:**
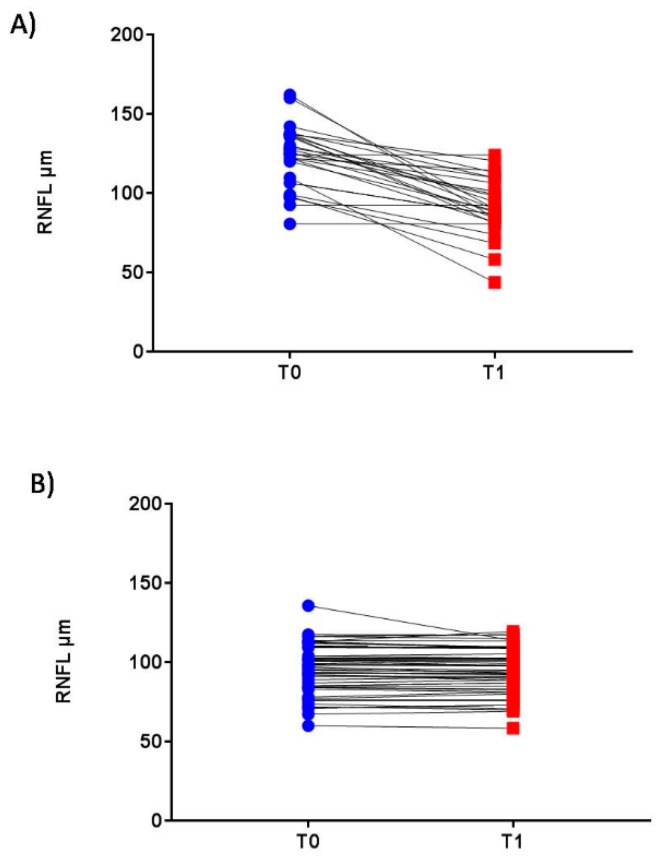
Longitudinal changes of the retinal nerve fiber layer in ON and nON eyes. Longitudinal changes of the retinal nerve fiber layer (RNFL) in optic neuritis (ON) and nonoptic neuritis (nON) eyes at T0 (acute phase of the relapse) and T1 (after 180 ± 2 days). (**A**) ON eye: RNFL measurement in the ON affected eye decrease in the retinal thickness in ON eyes, whereas the RNFL showed an initial increase with a subsequent decline. (**B**) nON eyes: RNFL measurement in the nonaffected eye showed a slight reduction at the follow-up. RNFL: retinal nerve fiber layer; ON: optic neuritis, nON: nonoptic neuritis, T0: acute phase of the relapse; T1: 180 ± 12 days after relapse onset. The reported data refer to the whole sample (*n* = 90).

**Figure 2 jcm-08-02022-f002:**
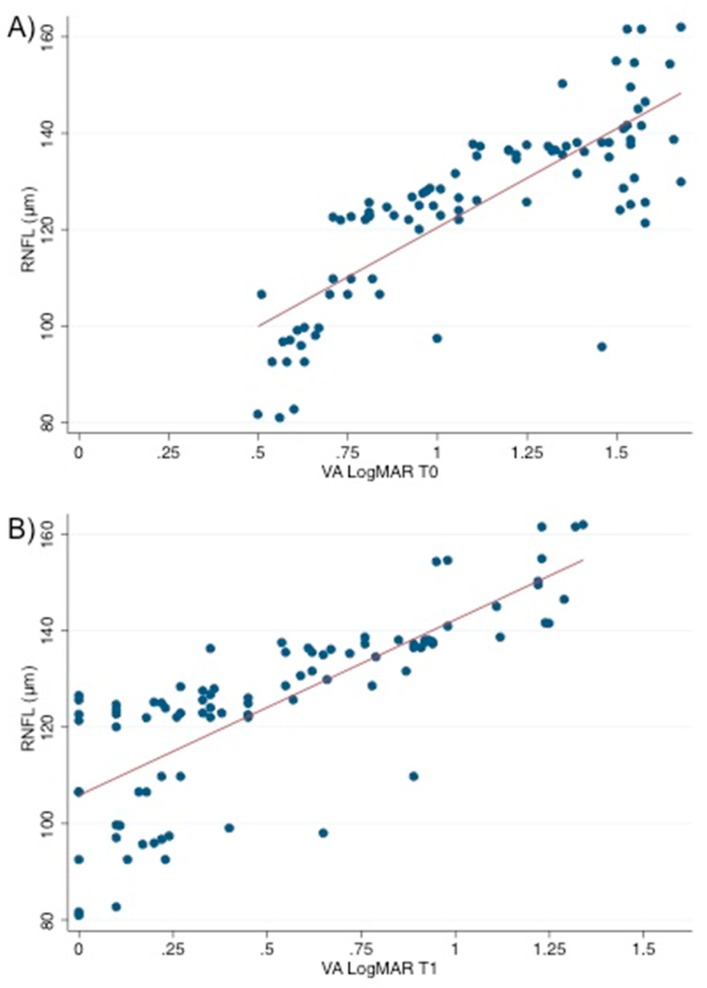
Correlation between retinal thickness and visual acuity in ON eyes. This figure shows the positive correlation (**A**) between retinal nerve fiber layer (RNFL) at T0 and visual acuity converted to the logarithm of the minimal angle of resolution (VA LogMAR) at T0 (*r* = 0.89; *p* < 0.0001) and (**B**) between retinal nerve fiber layer (RNFL) at T0 and visual acuity converted to the logarithm of the minimal angle of resolution (VA LogMAR) at T1 (r = 0.87; *p* < 0.005). RNFL: retinal nerve fiber layer; VA LogMAR: visual acuity converted to the logarithm of the minimal angle of resolution. The reported data refer to the whole sample (n = 90). The *p*-value obtained through Spearman’s correlation analysis is also shown.

**Table 1 jcm-08-02022-t001:** Demographic and clinical characteristics of ON MS patients.

	Total (90)
Female (%)	60 (66.7%)
Age (mean ± SD)	35.6 ± 10.5
Disease Duration (months; mean ± SD)	91.9 ± 85.2
Total N of relapses (mean ± SD)	5.3 ± 4.3
EDSS during relapse (mean ± SD)	3.1 ± 1.6
EDSS postrelapse (mean ± SD)	2.5 ± 1.8
ΔEDSS (mean ± SD)	0.45 ± 0.06
Duration of visual symptoms (hours; mean ± SD)	35.9 ± 4.8
N patient recovered (%)	47 (52.2%)

MS: multiple sclerosis; N: number; F: female, SD: standard deviation; ON: optic neuritis; EDSS: Expanded Disability Status Scale; ΔEDSS: difference between EDSS prerelapse and postrelapse.

**Table 2 jcm-08-02022-t002:** Differences in clinical and retinal data ^a^ between ON and nON eyes in the cohort of MS patients at baseline (T0).

	ON Eyes	nON Eyes	*p*-Value
RNFL (µm)	129.1 ± 19.5	100.5 ± 10.1	<0.001
Superior sector (µm)	141.6 ± 12.6	110.3 ± 8.1	<0.001
Inferior sector (µm)	136.3 ± 9.0	115.3 ± 7.9	<0.001
Nasal sector (µm)	81.6 ± 8.9	78.6 ± 10.3	<0.05
Temporal sector (µm)	86.3 ± 9.4	75.5 ± 8.4	<0.05
Macular Volume (µm)	9.3 ± 0.8	9.5 ± 0.6	ns
Foveal Thickness (µm)	243.4 ± 15.0	245.8 ± 16.3	ns
Ganglional Cell (µm)	80.4 ± 8.8	86.8 ± 11.6	<0.05
LogMAR VA	0.42 ± 0.23	0.05 ± 0.05	<0.001
LCSLA charts 100%	19.9 ± 9.9	36.5 ± 5.2	<0.001
LCSLA charts 2.5%	9.1 ± 6.8	26.4 ± 10.2	<0.001
LCSLA charts 1.25	4.6 ± 5.7	19.6 ± 8.4	<0.001

^a^ The reported data refer to the whole sample (*n* = 90) and are expressed as mean ± standard deviation. The *p*-values obtained through unpaired student’s *t*-test are also shown. ON: optic neuritis; nON: non optic neuritis; RNFL: retinal nerve fiber layer; LCSLA: low-contrast Sloan letter acuity charts (for 100%, 2.5%, and 1.25% intensity of contrast); ns: not significant.

**Table 3 jcm-08-02022-t003:** Differences in clinical and retinal data ^a^ between T0 and T1 in ON and nON eyes.

	ON		nON		
	T0	T1	T0	T1	*p*-Value
RNFL (µm)	129.1 ± 19.5	91.6 ± 20.2	100.5 ± 10.1	93.1 ± 15.2	* < 0.001
					# < 0.05
Superior sector (µm)	141.6 ± 12.6	104.9 ± 8.9	110.3 ± 8.1	109.6 ± 7.3	* < 0.001
Inferior sector (µm)	136.3 ± 9.0	110.6 ± 10.6	115.3 ± 7.9	114.9 ± 11.6	* < 0.001
Nasal sector (µm)	81.6 ± 8.9	70.9 ± 9.9	78.6 ± 10.3	70.6 ± 12.3	*# < 0.05
Temporal sector (µm)	86.3 ± 9.4	71.5 ± 10.3	75.5 ± 8.4	68.6 ± 6.7	*# < 0.05
Macular Volume (µm)	9.3 ± 0.8	8.6 ± 0.7	9.5 ± 0.6	9.3 ± 0.5	* < 0.05
Foveal Thickness (µm)	243.4 ± 15.0	240.4 ± 13.4	245.8 ± 16.3	243.9 ± 10.8	ns
Ganglional Cell (µm)	80.4 ± 8.8	73.8 ± 11.6	86.8 ± 11.6	82.4 ± 8.1	*# < 0.05
LogMAR VA	0.42 ± 0.23	0.05 ± 0.02	0.05 ± 0.05	0.03 ± 0.05	* < 0.001
LCSLA charts 100%	19.9 ± 9.9	40.4 ± 8.5	36.5 ± 5.2	36.0 ± 6.2	* < 0.001
LCSLA charts 2.5%	9.1 ± 6.8	29.5 ± 11.2	26.4 ± 10.2	26.1 ± 11.8	* < 0.001
LCSLA charts 1.25	4.6 ± 5.7	21.0 ± 9.1	19.6 ± 8.4	18.5 ± 8.7	* < 0.001

^a^ The reported data refer to the whole sample (*n* = 90) and are expressed as mean ± standard deviation. The *p*-values obtained through paired t-test are also shown. ON: optic neuritis; nON: non optic neuritis; RNFL: retinal nerve fiber layer; VA LogMAR: visual acuity converted to the logarithm of the minimal angle of resolution during relapse (best recorded values). LCSLA: low-contrast Sloan letter acuity charts (for 100%, 2.5%, and 1.25% intensity of contrast; ns. not significant. * T0 versus T1 inON eyes; # T0 versus T1 in nON eyes. ns: not significant.

**Table 4 jcm-08-02022-t004:** Mean intereye differences (values in nonaffected eyes minus affected eyes) in optical coherence tomography and visual function parameters in ON patients.

	T0	T1	*P* Value
RNFL	−29.6	5.5	<0.01
Ganglional Cell Layer	5.4	11.4	<0.01
Macular Volume	0.2	0.7	ns
Foveal Thickness	2.4	3.5	ns
LogMAR VA	−1.6	−0.4	<0.05

ON: optic neuritis; T0: acute phase; T1: 180 ± 12 days after relapse onset; RNFL: retinal nerve fiber layer; VA LogMAR: visual acuity converted to the logarithm of the minimal angle of resolution during relapse (best recorded values) for affected eye in ON patients and for the better eye in nON patients. ns: not significant. The reported data refer to the whole sample (n = 90). The *p*-values obtained through paired student’s t-test are also shown.

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
