# Peer review of "Retinal Nerve Fiber Layer Thickness and Higher Relapse Frequency May Predict Poor Recovery after Optic Neuritis in MS Patients"

_jcm, 2019, doi:10.3390/jcm8112022_

Round 1
Reviewer 1 Report
All my concerns and comments have been adressed sufficiently. I have no further comments.
Author Response
Dear Editor,
We are grateful to the reviewer for the insightful feedback and the opportunity to improve our manuscript.
Best regards,
The corresponding author
Reviewer 2 Report
This is a prospective, observational study looking at visual recovery after an attack of optic neuritis. The authors conclude that a greater number of prior non-ON MS attacks and increased optic disc swelling predict poor recovery after optic neuritis. In general the data for number of relapses appear sound, and the result is interesting. However, the use of quartiles to define grades of optic disc edema has created a circular description. This analysis needs to be approached in a more sensible way. Also, both the introduction and discussion are excessively wordy and hard to follow, and the entire manuscript needs some editing for English. More specific comments are outlined below.
Abstract:
No need to capitalize Multiple Sclerosis and Optical Coherence Tomography
Lines 40-41: It is not clear what is being compared here. Please revise. Perhaps ON eye at T1 vs nON eye at T1 would be a better data point to include here.
Introduction:
Line 57: suggest "biomarkers and therapies"
Line 65: extra comma, no need for "the" inflammation
Line 63-67: very long run on sentence
Line 74: suggest "allows follow-up of clinical evolution"
Materials and Methods
Line 91: "who experienced an ON MS relapse" The use of the word relapse makes it unclear whether these patients were having their first or second episode of optic neuritis.
Line 115: please specify dose and route of steroids. Was every patient treated with steroids, regardless of severity of the attack?
Line 127: I do not understand "partial recovery was equaled to non-recovery". Does this mean that partial recovery was considered the same as non recovery?
Line 129-134: no need to capitalize Optic Disc Cube, Macular Volume, Foveal Thickness etc
Line 149: was grade 0 derived from the nON eyes? Or just the lowest quartile of ON eyes? Is there a precedent for this grading scheme? Why not use the generally accepted Frisen grade?
Results
Line 223-225, Figure 1: If swelling grade was determined by quartile, of course there is going to be an even distribution of swelling grade. This is a completely useless parameter. What is the intended purpose of this analysis? Please eliminate figure 1 or use a better way to determine the amount of optic disc edema.
Line 264, Table 3: It would be useful to compare vs the fellow eye at T1 as well. Consider combining table 3 and 4.
Line 322 Figure 2: You contradict yourselves. Here you state that there was no significant change in RNFL measurements, yet in table 4 and in the text, you show a significant decrease in mean, nasal and temporal RNFL thickness.
Line 419 figure 3: I would be more interested to know how initial RNFL at T0 compared to VA at T1.
Discussion
Line 437-439: This sentence makes no sense, please revise.
Line 441: what is "a certain grade"?
Line 447: "are thought to be in charge of the maintenance of..." is not a very scientific explanation
The entire first paragraph of the discussion is very poorly written and hard to follow.
Line 459-461: This is an interesting point and should be elaborated upon. Does the fellow eye experience loss of RNFL and GCC continuously throughout the disease course of MS? Or does this only occur during an attack of optic neuritis? If the former, a mean loss of 7um over 6 months, with an initial thickness of 100um, would predict a complete loss of all RNFL in just over 7 years! If the latter, does this mean that an attack of unilateral optic neuritis is actually bilateral but asymmetric?
Line 466: Be careful what you say about RNFL swelling, since the way you defined it was circular.
Line 492: “…disability and previous relapses; although the different clinical profile, patients with previous ON…” This sentence (and others in the discussion) is grammatically incorrect, and therefore confusing. Please edit for English
Author Response
Dear Editor,
We are grateful to the reviewer for the insightful feedback and the opportunity to improve our manuscript. Some aspects were not clearly described and we thoroughly revised the paper to improve its transparency and quality.
Please find below a point by point answer to the reviewer’s comments. For each comment, we specified how the paper was modified (in bold and highlighted).
COMMENTS AND SUGGESTIONS FOR AUTHORS
This is a prospective, observational study looking at visual recovery after an attack of optic neuritis. The authors conclude that a greater number of prior non-ON MS attacks and increased optic disc swelling predict poor recovery after optic neuritis. In general the data for number of relapses appear sound, and the result is interesting. However, the use of quartiles to define grades of optic disc edema has created a circular description. This analysis needs to be approached in a more sensible way. Also, both the introduction and discussion are excessively wordy and hard to follow, and the entire manuscript needs some editing for English. More specific comments are outlined below.
- Thank you for these comments. The choice of quartiles instead of Frisen grade was related to the need to objectivise the swelling (see below in more detail). We tried to modify the incorrect sentences and edit the manuscript for English.
ABSTRACT:
No need to capitalize Multiple Sclerosis and Optical Coherence Tomography.
- We modified as suggested.
Lines 40-41: It is not clear what is being compared here. Please revise. Perhaps ON eye at T1 vs nON eye at T1 would be a better data point to include here.
- We modified this sentence as suggested (see pag 2, line 39).
INTRODUCTION:
Line 57: suggest "biomarkers and therapies".
- We corrected as indicated.
Line 65: extra comma, no need for "the" inflammation.
- We deleted the extra comma as suggested.
Line 63-67: very long run on sentence.
- We rephrase the sentence as suggested (pag 3, lines 60-63)
Line 74: suggest "allows follow-up of clinical evolution".
- We modified as suggested.
MATERIALS AND METHODS
Line 91: "who experienced an ON MS relapse" The use of the word relapse makes it unclear whether these patients were having their first or second episode of optic neuritis.
- We agree the Reviewer. We deleted the term “relapse”.
Line 115: please specify dose and route of steroids.
- Dose and route of steroids are reported in the Results section (pag. 5-6, lines 156-160).
Was every patient treated with steroids, regardless of severity of the attack?
- All patients were treated with steroids (methylprednisolone 1000 mg/die) intravenously administered for 5 days, as for protocol in our clinic.
Line 127: I do not understand "partial recovery was equaled to non-recovery". Does this mean that partial recovery was considered the same as non recovery?
- We confirm that we considered partial recovery as no recovery.
Line 129-134: no need to capitalize Optic Disc Cube, Macular Volume, Foveal Thickness etc.
- We corrected as suggested.
- Line 149: was grade 0 derived from the nON eyes?Or just the lowest quartile of ON eyes?
- The grade derived from ON eyes. In order to stratify the RNFL swelling severity, we used 25% percentiles and median values as explained in Statistical Analysis section.
Is there a precedent for this grading scheme? Why not use the generally accepted Frisen grade?
- To the basis of our knowledge, this grading was not previously used. However, we did not use the Frisen grade in order to objectivize the optic disk evaluation reducing the intra-rater variability. This was stated in the discussion section (pag 13, lines 426-429).
RESULTS
Line 223-225, Figure 1: If swelling grade was determined by quartile, of course there is going to be an even distribution of swelling grade. This is a completely useless parameter. What is the intended purpose of this analysis? Please eliminate figure 1 or use a better way to determine the amount of optic disc edema.
- We thank the Reviewer for the comment. We deleted the figure 1 as suggested.
- Line 264, Table 3: It would be useful to compare vs the fellow eye at T1 as well. Consider combining table 3 and 4.
- We thank the Reviewer for the suggestion. We combined tables 3 and 4.
- Line 322 Figure 2: You contradict yourselves. Here you state that there was no significant change in RNFL measurements, yet in table 4 and in the text, you show a significant decrease in mean, nasal and temporal RNFL thickness.
- We apologize for the typo. We corrected in Figure 2.
- Line 419 figure 3: I would be more interested to know how initial RNFL at T0 compared to VA at T1.
-We added a correlation analysis between RNFL at T0 and VA at T1 as suggested.
DISCUSSION
Line 437-439: This sentence makes no sense, please revise.
- We refreshed the sentence as suggested.
- Line 441: what is "a certain grade"?
- We modified as indicated.
Line 447: "are thought to be in charge of the maintenance of..." is not a very scientific explanation
The entire first paragraph of the discussion is very poorly written and hard to follow.
- We rewrote the paragraph as indicated.
- Line 459-461: This is an interesting point and should be elaborated upon. Does the fellow eye experience loss of RNFL and GCC continuously throughout the disease course of MS? Or does this only occur during an attack of optic neuritis? If the former, a mean loss of 7um over 6 months, with an initial thickness of 100um, would predict a complete loss of all RNFL in just over 7 years! If the latter, does this mean that an attack of unilateral optic neuritis is actually bilateral but asymmetric?
- We thank the Reviewer for the comment. Many studies demonstrated that MS is associated with longitudinal thinning affecting RNFL and GCL thickness. Particularly, prior optic neuritis (together with female gender, disease duration etc) was one of predictive factor of faster rate of neuro-axonal loss, not only in the affected eye but also in the fellow eye (see Lisanne J Balk et al. Mult Scler J Exp Transl Clin. 2019 Sep 5;5(3):2055217319871582 and Behbehani R et al. Mult Scler Relat Disord. 2018 Apr;21:56-62.). However, this progressive retinal thinning does not show a constant rate of reduction. Instead, significant progressive neuronal and axonal degeneration is dependent on disease duration and it is greatest in early phases of the MS disease where the pro-inflammatory process is more pronounced than the degenerative one (see Lisanne J. Balk et al. J Neurol (2016) 263:1323–1331).
Line 466: Be careful what you say about RNFL swelling, since the way you defined it was circular.
- We modified this where needed.
- Line 492: “…disability and previous relapses; although the different clinical profile, patients with previous ON…” This sentence (and others in the discussion) is grammatically incorrect, and therefore confusing. Please edit for English.
- We rephrased the incorrect sentences and revised all the manuscript for English in order to improve the quality of the paper.
